# Longitudinal Association of Economic Growth with Lung Function of Chinese Children and Adolescents over 30 Years: Evidence from Seven Successive National Surveys

**DOI:** 10.3390/ijerph18126592

**Published:** 2021-06-19

**Authors:** Xiaomei Gan, Kehong Yu, Xu Wen, Yijuan Lu

**Affiliations:** 1Department of Sport Science, College of Education, Zhejiang University, Hangzhou 310000, China; 11603020@zju.edu.cn (X.G.); wenxu@zju.edu.cn (X.W.); 11703023@zju.edu.cn (Y.L.); 2Center for Sports Modernization and Development, Zhejiang University, Hangzhou 310000, China

**Keywords:** economic growth, lung function, longitudinal association, children and adolescents

## Abstract

(1) Background: Recent studies reported that decrease in lung function of Chinese children and adolescents continues to decline, although the change has been insignificant and has reached a plateau. However, studies have not explored the relationship between lung function and economic development in China. This study sought to explore the longitudinal association between socio-economic indicators and lung function; (2) Method: Data were obtained from seven successive national surveys conducted by the Chinese National Survey on Students’ Constitution and Health from 1985 to 2014. Lung function of school-age children (7–22 years) was determined using forced vital capacity (FVC). GDP per capita and urbanization ratio were used as economic indicators. A fixed-effects model was employed to examine the longitudinal association after adjusting for height, weight, and time trends; (3) Results: Socio-economic indicators showed a U-curve relationship with lung function of boys and girls from urban and rural areas. Lung function initially decreased with GDP per capita or urbanization ratio and reached a minimum. Lung function then increased with increase in GDP per capita or urbanization ratio. The findings indicate that the relationship between economic growth and lung function is different in different development stages. In less-developed provinces, economic growth was negatively correlated with lung function, whereas, in developed provinces, economic growth was positively correlated with lung function; (4) Conclusion: The findings of the current study show that economic growth has significantly different effects on lung function at different economic levels. Therefore, governments should improve lung health in children and adolescents from low and middle economic regions.

## 1. Introduction

Poor lung function is a predictor of respiratory diseases, cancer, cardiovascular diseases, and all-cause mortalities [1,2]. Studies report that lifestyle-related chronic diseases accounted for two-thirds of all global deaths between 2002 and 2030 [3], causing significant economic losses [4]. Lung function depends on several biological factors such as gender, height, weight, and genetic factors [5,6,7]. However, it is significantly affected by socio-economic and environmental factors, such as ethnicity, air pollution, nutrition, physical activity, sedentary behavior, and socioeconomic status (SES) [4,7,8,9,10], which are highly correlated with macro-economy levels.

Despite the significant effects of economic development on lung function, studies have not fully explored the relationship between children’s lung function and economic growth. Several studies have mainly focused on the association between individual-level indicator (SES) and respiratory function [7,9,11,12]. Socio-economic status does not adequately explain vital capacity [7], because SES does not cover all economic factors. Furthermore, these studies mainly used a cross-sectional design, limiting the correct inferences on causality [11].

China is a rapidly industrializing nation [13] whose economy has grown rapidly over the past 30 years, however, this growth is associated with unique environmental hazards and significant respiratory disease burden [14]. Notably, only a few studies have explored the effects of economic growth on lung function of children and adolescents over the last 30 years. A national survey conducted from 1984 to 2005 reported continuous decline in FVC of students (7–22 years). Furthermore, a continuous decline in FVC was reported in 2010 [15]. However, studies have not fully explored why the turning point occurs. Similar to the relationship between income inequality and development [16], the concept of the Kuznets curve relationship has been recently applied to explore personal health issues [17]. For instance, previous studies reported an obesity Kuznets curve [18,19] and a cardiorespiratory fitness Kuznets curve [20]. To predict lung function and provide empirical evidence for formulating public health policy, it is important to explore whether there is a Kuznets curve relationship between lung health and economic growth. Currently, no study has explored the association of economic growth with lung function in children and adolescents.

In the current study, data from 1.5 million children and adolescents aged between 7 and 22 years were collected from 27 provinces in China between 1985 and 2014. Data were used to explore lung function status in children and adolescents and its relationship with economic development, height, and weight. The relationship between economic development and lung function was explored after adjusting for anthropometric, provincial fixed effects, time-specific effects, and other factors. The current study provides empirical evidence on the relationship between the economy and children’s lung function in China. These findings are important for policy formulation and development of effective interventions. The findings for the study serve as a reference for similar studies in other developing countries.

## 2. Methods

### 2.1. Study Design and Participants

Data were obtained from seven comprehensive national successive surveys conducted by the Chinese National Survey on Students’ Constitution and Health (CNSSCH) in 1985, 1991, 1995, 2000, 2005, 2010, and 2014. Participants were students aged 7–22 years, selected from specific areas in each province between 1985 and 2014. Participants were selected using a stratified cluster sampling approach from certain classes. Clusters were randomly selected from each grade in the selected schools. Informed consent was obtained from children and their parents before carrying out the study. CNSSCH procedures used were the same during the 30 years [15].

The study included data of respondents from 27 out of the 31 provinces in China. Hainan, Chongqing, Qinghai, and Tibet provinces were excluded from the study. Hainan and Chongqing were founded after 1985, whereas Qinghai was not included in the 1995 survey. Tibet is an autonomous province that was not covered in nearly all surveys.

### 2.2. Measurement

All seven CNSSCH surveys underwent a complete anthropometric evaluation following a standard protocol in all survey sites. Forced vital capacity (FVC) was used to reflect lung function of participants [21]. Height (cm), weight (kg), and FVC (ml) were determined by a team of trained technicians following a standardized procedure. Height was measured to the nearest 0.1 cm using portable stadiometers, whereas weight was measured to the nearest 0.1 kg using a standardized scale and expressed as a mean of three measurements. A rotary spirometer was used to determine FVC in 1985 and 1991 surveys, whereas an electronic spirometer was used from 1995 to 2014. Most students took all tests on the same day. The sample sizes and participants proportion in each test item are presented in Table 1.

Real per capita GDP and urbanization ratio at national level for each survey year was used to determine the level of economic growth which was derived from China’s statistical yearbooks of the National and Provincial Bureau of Statistics of China. These yearbooks provide authoritative data for real GDP per capita for each province, which are aggregated to represent national macroeconomic growth adjusted for purchasing power parity exchange rates, thus making the growth of GDP per capita comparable.

### 2.3. Statistical Analysis

All statistical analyses were performed using Stata version 16 software. Descriptive statistics were used for analysis of demographic information for each survey year and the trends in FVC, height, and weight from 1985 to 2014. A panel econometric model was used to estimate the relationship between socio-economic indicators and lung function. The panel structure allowed control for time-invariant province characteristics such as geographical location. These characteristics were expected to be correlated with socio-economic indicators. Therefore, a fixed-effect panel econometric model was used to control the province-level particularities and the time trends [22]. Models that exhibited a linear, quadratic relationship between socio-economic indicators and respiratory function were developed to obtain one that could accurately represent the data (models are shown in Appendix A). Models were developed after adjusting for several covariates and confounders, including height, weight, and time effect. Non-stationarity of the variables was accounted for and the growth rate of GDP per capita in log differences was used to robust the results in linear and quadratic models (shown in Appendix B).

Robust command was used to correct the standard error with white heteroscedasticity based on the natural logarithm of each variable to ensure robust results. U-Curve hypothesis was tested by three steps in the estimation of quadratic model [23]. Initial tests explored whether β_1_ < 0 and β_2_ > 0 and if both were statistically significant. Estimations on two subsamples (the males and females) were then conducted. Further, analysis was performed to explore whether the turning point at which the curve attained its minimum was located accurately within the data range. The point was calculated as: *τ* = −β_1_/2β_2_. To further observe the trajectory of associations of economic growth with lung function, the fixed effect model was used to test associations for the total sample and separately by sex, region within 10 years (1985–1995, 1995–2005, 2005–2014), after adjusting for age, BMI, provincial and time specific effects, and any time-varying differences common to all provinces.

In addition to several checks such as standard two-way fixed effects estimation, exclusion of non-stationarity of variables, U-Curve hypothesis test, and periodical examination to determine the robustness of the results and specification errors, instrumental variable (IV) estimation—which uses infant mortality as the instrument for GDP per capita—was used to simultaneously cope with endogeneity and panel non-stationarity (shown in Appendix C).

## 3. Results

### 3.1. Trends Results

Characteristics of the study population included in the seven surveys are presented in Table 1. National trends in FVC performance were curvilinear over time, with a gradual decline from 1985 to 2005 and an improved directional shift from 2005 to 2014 (Figure 1). Students showed the worst respiratory function in 2005. Mean height and weight increased with increase in GDP per capita (Table 1).

Mean FVC in urban boys and girls decreased from 3063.6 (SD 1077.1) in 1985 to 2753.7 (SD 1022.7) in 2005 and then increased to 2982.3 (SD 1077.3) in 2014. This trend was consistent for both genders and different age groups in both urban and rural areas. FVC decline in girls was significantly higher in both urban and rural areas from 1985 to 2005 compared with that of boys.

### 3.2. Linear Analysis

The performance of linear specification between socio-economic indicators in urban and rural areas and lung function of children and adolescents after adjusting for height, weight, age time effect, and fixed effects are presented in Table 2 (columns labeled ‘linear’). GDPPC models showed significant negative coefficients (−0.022 and −0.039, and−0.032 and −0.056) for boys and girls from urban and rural areas, respectively. Coefficients represented the elasticity of FVC to GDP per capita based on the log-log specification. A 1% increase in the GDP per capita was correlated with a decrease in FVC by 0.022% and 0.039% for urban boys and girls, respectively. For example, a 100% (2224 US$) increase in the mean value of PCDI (mean = 2224 US$) caused a 2.2% (65 mL) decrease in the mean value of FVC (mean = 2951 mL) in urban boys (*p* < 0.01).

This finding shows that urbanization ratio had significantly negative linear association with lung function in boys and girls above 30 years old. These findings further show that economic growth had a significantly negative effect on the respiratory function of children and adolescents in both urban and rural areas. Economic growth was associated with significant decrease in lung function in rural children and adolescents compared with their urban peers. In both urban and rural areas, the negative effect of economy on lung capacity of female students was significantly higher compared with that of male students.

### 3.3. Quadratic Analysis

Analysis using linear models showed a negative correlation between economic growth and lung function, which was inconsistent with the descriptive results. A quadratic model was established to obtain a specification that best represents the data. The quadratic specification results are presented in Table 2 (columns labeled ‘quadratic’). The coefficient on the log of GDP per capita was significantly negative, whereas that on the squared log of GDP per capita was significantly positive for urban full sample (β_1_ = −0.866 (*p* < 0.01); β_2_ = 0.057 (*p* < 0.01)) and rural full sample (β_1_ = −0.575 (*p* < 0.01); β_2_ = 0.036 (*p* < 0.01)). The turning points (exp(−β_1_/2β_2_)) were at 1978 US$ per capita for the urban full sample and 2921 US$ for the rural full sample. The turning points τ, that is, the minimum point of the U-curve for urban and rural samples, occurred between 2005 and 2010, as shown in Table 1.

Analysis of urbanization ratio indicated that β_1_ > 0 and β_2_ < 0, were statistically significant, implying that urbanization ratio exhibited a nonlinear relationship with lung function in all samples. In addition, turning points τ in the U- curve of urbanization ratio with lung function occurred between 2005 and 2010. In urban and rural areas, the estimates based on male and female student samples exhibited a “U” curve (Figure 2 and Figure 3). The turning point in all models for rural students occurred in later years, compared with models for urban peers.

### 3.4. Periodical Analysis

To further observe the trajectory of associations, associations for the total sample and separately by sex, region within 10 years (1985–1995, 1995–2005, 2005–2014) were explored (Figure 4 and Figure 5). For the whole sample, each 1-standard-deviation increase in GDPPC was associated with a 1.27-standard-deviation decrease in lung function of students from 1985–1995. Moreover each 1-standard-deviation increase in GDPPC was correlated with a 0.84-standard-deviation in decrease of in lung function of students from 1995–2005. The negative impact in 1995–2005 was significantly less compared with that from 1985–1995. On the contrary, economic growth was significantly positively correlated with lung function for all sample from 2005–2014. The findings on analyses of urban, rural, boys’ and girls’ samples were similar to those of the analyses of the whole sample. Findings on standardized association of urbanization ratio with lung function at each period were similar with the GDPPC findings. A significantly negative effect of urbanization level was observed between 1985–1995 and 1995–2005, whereas a positive effect of urbanization level was observed between 2005–2014.

Analysis showed a U-shaped relationship between economic growth and lung function by precisely representing the data of each sample. Lung function of students initially decreased with GDP per capita or urbanization ratio until it reached a minimum, after which lung function increased with increase in GDP per capita or urbanization ratio. These findings indicated that the relationship between economic growth and lung function varied in different development stages. Notably, in less-developed provinces, economic growth was negatively correlated with lung function whereas in developed provinces economic growth was positively correlated with lung function.

All regressions shown are adjusted for BMI, age, and provincial and time-specific effects and any time-varying differences common to all provinces. t statistics is shown in parentheses. GDP = gross domestic product.

All regressions shown are adjusted for BMI, age, and provincial and time-specific effects and any time-varying differences common to all provinces. t statistics is shown in parentheses.

## 4. Discussion

The findings of the study show a U-curve relationship between economic growth and lung function of children and adolescents. Lung function of students initially decreased with decrease in socio-economic indicators and then increased with increase in socio-economic indicators. These findings are consistent with the FVC report from 1984 to 2014, which shows a U-shaped trajectory [15] (Figure 1). In addition, the findings were consistent with longitudinal analyses between SES and lung function [24]. Analysis showed a decrease in the lung function of participants in the lowest childhood SES group. The highest group shows increased growth, whereas the middle group shows intermediate growth [24]. Although several studies report that the socio-economic circumstances are positively correlated with lung function [12,25,26,27], most of these studies used cross-sectional designs and only explored the effect at the individual level. A previous study reported an inverse correlation between SES and lung function. The study reported that the association between SES and lung function is correlated with the area’s economic level [28].

The current study presents several plausible explanations for the FVC–Kuznets curve pattern between economic growth and lung function in Chinese children and adolescents. China has experienced rapid economic development and urbanization since the implementation of the “Reform and Opening-up” policy in 1978. Studies report significant commuter exposures, air pollution, and construction pollutants in the environment [29]. Several studies report that air pollution is associated with a decline in respiratory function and a higher risk of respiratory illnesses [30,31,32]. Physical inactivity and sedentary lifestyles due to rapid development of the social economy and urbanization are also associated with decrease in lung function [8]. In addition, a sedentary lifestyle is highly associated with low respiratory muscle strength and breathing difficulties [33].

The government and society have taken measures to protect the environment even with socio-economic development [29]. Environmental policies have accelerated the arrival of the turning point for the whole country despite the low likelihood of the Environment–Kuznets curve turning points being reached soon [34]. China is now entering an important stage of increased urbanization by optimizing urban landscapes and forms [29], despite the air quality getting worse in cities with higher urbanization levels [35]. It is postulated that residing in greener areas is positively associated with better lung function [30].

Studies report a positive association between physical activity, physical fitness, and lung capacity [10,36]. Physical exercise controls several crucial elements of aerobic conditions. Forceful inhalation and deflation of the lungs strengthens respiratory muscles [36]. The endurance running level of Chinese urban students between 1984 and 2014 showed a U-shaped trajectory [20]. In addition, a U-shaped relationship was reported between economic growth and endurance running of children and adolescents between 1985 and 2014 [20]. Furthermore, dietary factors are potential mediators of the association between economic growth and lung function [37]. The nutritional status of an ethnically similar population is associated with significant differences in lung function during childhood [38]. Improving nutrition during growth and development can increase maximal lung function and decrease the risk of adult lung diseases [39].

To the best of our knowledge, this is the first study to explore the longitudinal relationship between economic growth and respiratory function of children and adolescents using a large sample size. In the current study, lung function trends were predicted after adjusting for a series of factors, including height, weight, and time trends. The study period coincides with the 30 years of China’s reform and opening up, during which significant unbalanced changes of the socio-economic status occurred. The unbalanced development has led to several health concerns and inequalities in social welfare and income. Health characteristics of children and adolescents have also changed with the increase in socio-economic development. This study provides a basis for governments to formulate policies for distributing welfare payments and providing health education and health services. Moreover, it provides a basis for developing specific interventions that target priority populations, such as children and adolescents from areas with middle and low economies. The study further provides a basis for formulation of policies in other developing countries, including the value of using multiple indicators for tracking social and economic determinants of the respiratory function in children and adolescents.

Although the study has several strengths, such as using tracking data across 30 years spanning major economic change, use of large sample size across multiple provinces in China, and inclusion of several robustness checks, it had a few limitations. First, FVC was determined using different instruments across the 30 years. However, this did not significantly affect the assessment of respiratory function trends over time, mainly over the seven survey cycles. Second, the lung function indicator was specific. However, the main purpose of the study was to track lung function trends. Third, socioeconomic indicators were calculated based on provincial-level data, rather than average individual-level data. However, province-level socioeconomic inequalities reflect the effects of macroeconomic change on individual health outcomes of children and adolescents [40]. Fourth, data on other factors such as air pollution, physical activity, and socioeconomic status were not collected, which might substantially affect changing trends of children’s lung function. However, these factors might be mediators of macroeconomic effects on lung function and, therefore, do not require moderation [22]. Finally, the study did not explore the effect of environmental protection laws on the trends of lung function. However, this has little effect on the robustness of the results since several legal regulations were controlled using the two-way fixed effects estimation.

## 5. Conclusions

In summary, the national trends in lung function of Chinese children and adolescents exhibited curvilinear characteristic over time, with a gradual decline from 1985 to 2005 and an improved directional shift from 2005 to 2014. The current study showed the effect of economic growth on lung function in children and adolescents using panel fixed effects model. In addition, it explored the longitudinal association between economic growth and lung function using polynomial (quadratic) models and periodical regression. These findings show that economic growth exhibits a U-shaped relationship with lung function in youth over 30 years. This indicates that the association between economic growth and lung function varied in different development stages. In the low economic development stage, economic improvement causes a decrease in lung function levels. On the contrary, economic growth showed an increase in lung function in highly developed areas. The findings of this study provide a basis for governments to formulate policies on distributing welfare payments and providing health education and health services. Moreover, it provides specific interventions that target priority populations, such as children and adolescents from areas with middle and low economies.

## Figures and Tables

**Figure 1 ijerph-18-06592-f001:**
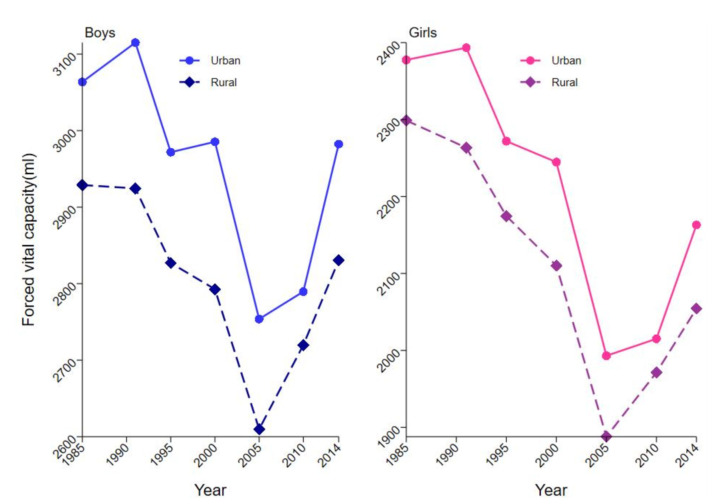
Trends in lung function indicators from 1985 to 2014.

**Figure 2 ijerph-18-06592-f002:**
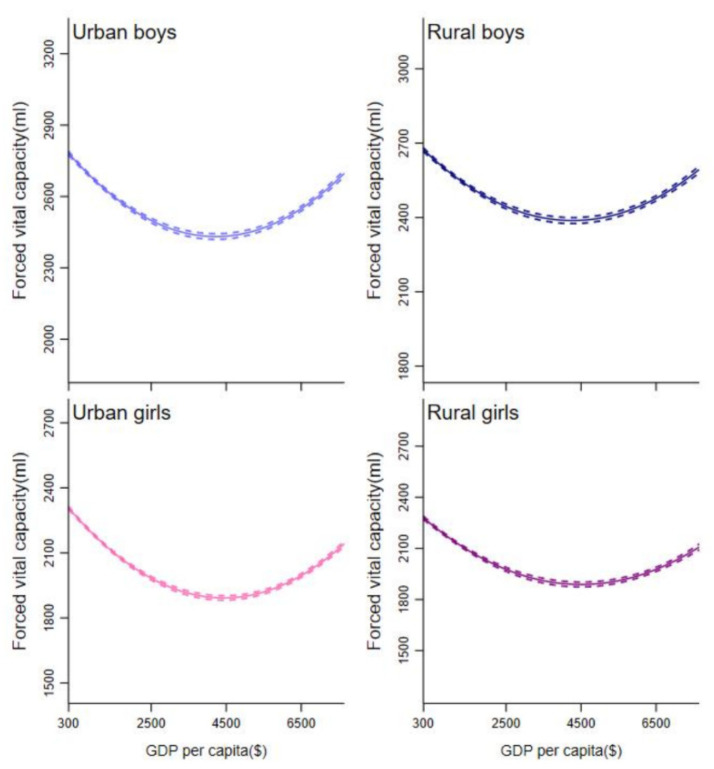
Association between per capital GDP and lung function from 1985 to 2014. Separate models were fitted for lung function by adjusting for age, height, weight, time trends and other time-invariant characteristics. Dotted lines indicate 95% CIs. GDP = gross domestic product.

**Figure 3 ijerph-18-06592-f003:**
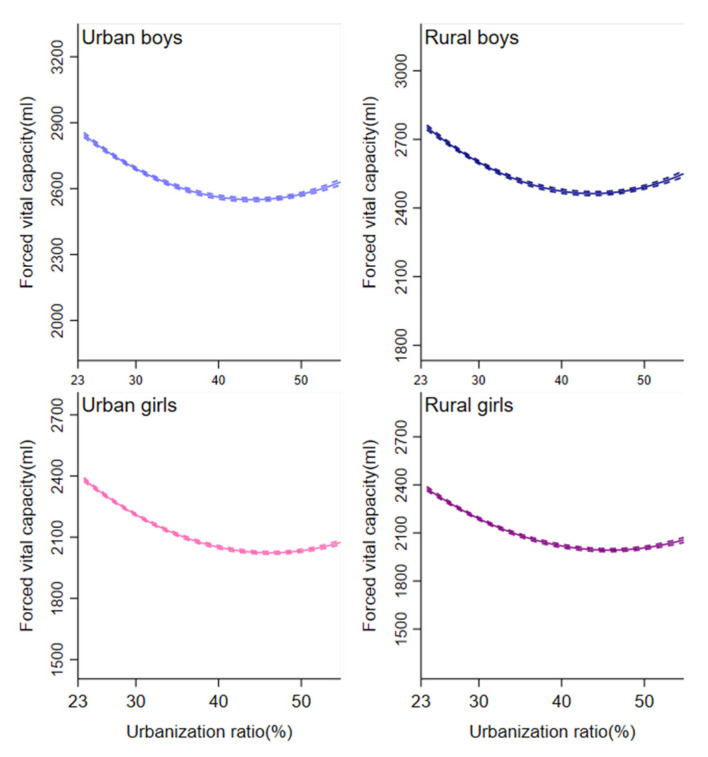
Association between urbanization ratio and lung function from 1985 to 2014. Separate models were fitted for lung function by adjusting for age, height, weight, time trends and other time-invariant characteristics. Dotted lines indicate 95% CIs.

**Figure 4 ijerph-18-06592-f004:**
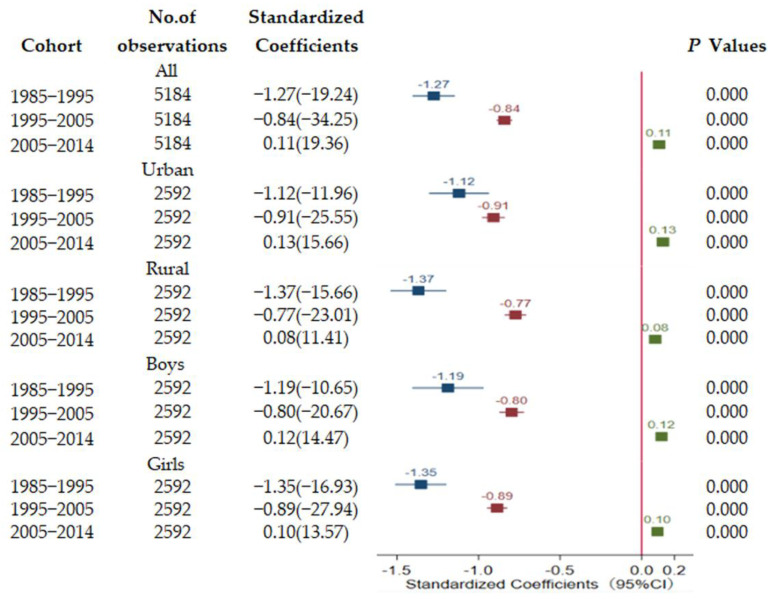
Longitudinal association of per capital GDP with lung function.

**Figure 5 ijerph-18-06592-f005:**
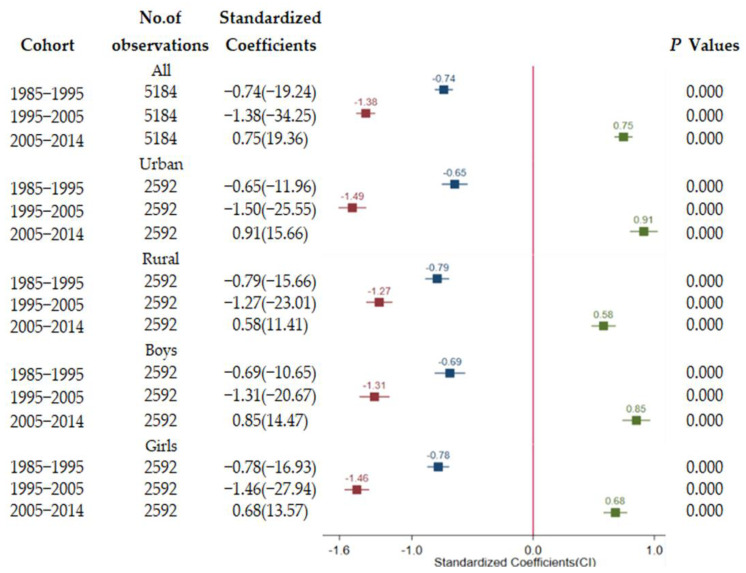
Longitudinal association of urbanization ratio with lung function.

**Table 1 ijerph-18-06592-t001:** Data for different survey years from the Chinese National Survey on Students’ Constitution and Health.

	1985	1991	1995	2000	2005	2010	2014
Sample size	409,836	184,835	204,763	216,073	234,289	215,223	214,301
Setting							
Urban	204,669 (49.9%)	93,545 (50.6%)	103,674 (50.6%)	108,564 (50.2%)	117,932 (50.3%)	107,537 (50.0%)	107,211 (50.0%)
Rural	205,167 (50.1%)	91,290 (49.4%)	101,089 (49.4%)	107,509 (49.8%)	116,357 (49.7%)	107,686 (50.0%)	107,090 (50.0%)
Sex							
Boys	205,040 (50.0%)	93,587 (50.6%)	102,998 (50.3%)	107,985 (50.0%)	117,594 (50.2%)	107,611 (50.0%)	107,175 (50.0%)
Girls	204,796 (50.0%)	91,248 (49.4%)	101,765 (49.7%)	108,088 (50.0%)	116,695 (49.8%)	107,612 (50.0%)	107,126 (50.0%)
FVC (mL)							
Male (urban)	3063.6 (1077.1)	3115.0 (1046.4)	2971.7 (1012.5)	2985.4 (1012.2)	2753.7 (1022.7)	2789.6 (1055.7)	2982.3 (1077.3)
Female (urban)	2377.4 (594.9)	2393.4 (564.8)	2271.9 (567.3)	2244.7 (553.6)	1993.0 (547.4)	2015.1 (569.2)	2163.1 (573.2)
Male (rural)	2928.9 (1082.7)	2924.5 (1076.6)	2826.9 (1020.2)	2792.8 (1008.6)	2609.7 (1006.4)	2719.7 (1049.9)	2830.4 (1063.9)
Female (rural)	2298.8 (632.5)	2263.3 (596.1)	2174.6 (600.2)	2110.2 (563.7)	1887.9 (551.1)	1971.3 (560.5)	2054.2 (572.4)
Height (cm)							
Male (urban)	154.1 (17.6)	156.0 (16.6)	156.6 (16.7)	157.5 (16.6)	158.4 (16.3)	159.4 (16.1)	160.3 (15.8)
Female (urban)	148.7 (13.1)	150.2 (12.3)	150.4 (12.2)	151.0 (12.2)	151.7 (11.9)	152.5 (11.6)	153.1 (11.3)
Male (rural)	150.4 (18.0)	152.5 (17.4)	153.2 (17.2)	154.1 (17.3)	155.5 (16.8)	156.5 (16.6)	158.1 (16.3)
Female (rural)	145.4 (13.9)	146.9 (13.2)	147.7 (12.9)	148.3 (12.9)	149.3 (12.4)	150.5 (12.5)	151.3 (11.8)
Weight (kg)							
Male (urban)	43.5 (13.6)	45.8 (13.2)	47.0 (13.5)	49.1 (14.0)	50.5 (13.5)	52.0 (13.7)	53.7 (13.9)
Female (urban)	40.2 (10.7)	41.4 (10.1)	42.3 (10.3)	43.4 (10.3)	44.1 (9.9)	44.9 (9.7)	46.2 (9.8)
Male (rural)	41.9 (13.9)	43.3 (13.8)	44.1 (13.8)	45.1 (13.9)	46.5 (13.5)	48.2 (13.6)	50.6 (13.6)
Female (rural)	39.6 (11.6)	40.2 (11.1)	41.0 (11.1)	41.4 (11.0)	42.1 (10.4)	43.4 (10.4)	44.5 (9.9)
GDP per capita (US$)	280	341	519	939	1669	4295	7529
Urbanization ratio	23.7%	26.4%	29%	36.2%	43.0%	48.0%	54.8%

Data are n, n (%), mean (SD). FVC, forced vital capacity; GDP, gross domestic product. GDP per capita is total GDP ÷ total population.

**Table 2 ijerph-18-06592-t002:** Econometric Estimates of the Relationship between Socio-economic indicators and Lung Function of Students.

	LFVC								
		Urban Boys		Urban Girls		Rural Boys		Rural Girls	
Models	Variable	*β*	*p* Values	*β*	*p* Values	*β*	*p* Values	*β*	*p* Values
Linear	LGDPPC	−0.02 (−6.75)	0.000	−0.04 (−14.08)	0.000	−0.03 (−11.38)	0.000	−0.06 (−21.38)	0.000
Quadratic	LGDPPC	−0.78 (−17.08)	0.000	−0.94 (−18.57)	0.000	−0.49 (−11.26)	0.000	−0.67 (−13.93)	0.000
	LGDPPC^^2^	0.05 (16.57)	0.000	0.06 (17.62)	0.000	0.03 (10.50)	0.000	0.04 (12.68)	0.000
		τ = 1669		τ = 1919		τ = 2921		τ = 2697	
Linear	LURBAN	−0.09 (−6.75)	0.000	−0.15 (−14.08)	0.000	−0.12 (−11.38)	0.000	−0.22 (−21.38)	0.000
Quadratic	LURBAN	−5.91 (−16.84)	0.000	−7.00 (−18.10)	0.000	−3.61 (−10.88)	0.000	−4.92 (−13.31)	0.000
	LURBAN^^2^	0.81 (16.57)	0.000	0.96 (17.61)	0.000	0.49 (10.49)	0.000	0.66 (12.66)	0.000
		τ = 42.94		τ = 42.09		τ = 44.70		τ = 50.90	

LFVC, log of forced vital capacity; LGDPPC, log of GDP per capita; LGDPPC^^2^, the squared term of LGDPPC. LURBAN, log of urbanization ratio; LURBAN^^2^, the squared term of LURBAN. τ is the turning point of “U”curve. White (1980) robust regression.t statistics is shown in parentheses. All regression shown are adjusted for height, weight, age and for provincial and time-specific effects and any time-varying differences common to all provinces.

## Data Availability

Datasets generated and analyzed during the current study are not publicly available due to participant confidentiality but are available from the corresponding author on reasonable request.

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
