# Peer review of "Longitudinal Association of Economic Growth with Lung Function of Chinese Children and Adolescents over 30 Years: Evidence from Seven Successive National Surveys"

_ijerph, 2021, doi:10.3390/ijerph18126592_

Round 1
Reviewer 1 Report
The paper provides an interesting point of view of the lung function issue of Chinese children and adolescents with the relation to the economic development.
The statistical data (especially research sample) and statistical analysis are particularly noteworthy.
Despite this, few changes are required for evaluating a possible publication:
Major comments: in the title you mention about the economic development, however in the text you write very often about economic growth. It’s not the same. You use the GDP per capita as an indicator of economic development which is debatable.
Economic development includes qualitative and quantitative factors. You have chosen only one (quantitative).
I suggest you to change the title: Longitudinal Association of Economic Growth with Lung Function of Chinese Children and Adolescents over 30 years: Evidence from Seven Successive National Surveys. Especially that in the table 2 you compare economic growth and lung function of students. The same as Figure 2.
If not… please write more about research limitations.
You write very little about economic development concept. You should in the introduction write more about it.
Besides you sometimes you equate GDP per capita to income. However, GDP per capita is not the same as average income because Gross domestic product measures how much every individual has contributed to the production. In contrast, per capita income measures the average income of every individual in the country.
In Discussion part you refer to Kuznets curve. You should in the introduction develop it.
The conclusion part is very modest. Please develop it.
Minor comments:
Line 30, square bracket: you should replace [1] [2] to [1-2]
Line 97 too large space between: … function. The
All references need thorough change according to the: Instructions for Authors.
Author Response
Dear reviewer,
Thank you for providing the detailed and constructive suggestions for our manuscript. We have carefully reviewed, addressed each comment below and made corresponding changes in the manuscript (highlighted in yellow).
We apologize that the writing of the manuscript in the previous version was not clear and caused confusion. The English of the manuscript was edited by a professional company.
We hope you find the changes satisfactory.
Sincerely
Xiaomei Gan
Major comments
Comment 1: in the title you mention about the economic development, however in the text you write very often about economic growth. It’s not the same. You use the GDP per capita as an indicator of economic development which is debatable. Economic development includes qualitative and quantitative factors. You have chosen only one (quantitative). I suggest you to change the title: Longitudinal Association of Economic Growth with Lung Function of Chinese Children and Adolescents over 30 years: Evidence from Seven Successive National Surveys. Especially that in the table 2 you compare economic growth and lung function of students. The same as Figure 2.
Response
Thank you for constructive suggestions.
The title was revised as your suggestion ( line 1).
Except for GDP per capita, we added urbanization ratio as one of socio-economic indicators, and explore the association of urbanization ratio with lung function. Revision was made in the manuscript (line 183-185, line 205-208, Table2, Figure 3 and 5).
Comment 2: Besides you sometimes you equate GDP per capita to income. However, GDP per capita is not the same as average income because Gross domestic product measures how much every individual has contributed to the production. In contrast, per capita income measures the average income of every individual in the country.
Response
We apologize for the confusing description .
Revision was made in the manuscript (line 28,178,193,198,232,327,329).
Comment 3: In discussion part you refer to Kuznets curve. You should in the introduction develop it.
Response
Thank you for constructive suggestions.
Revision was made in the manuscript (line 64-72).
Comment 4: The conclusion part is very modest .Please develop it.
Response
Thank you for constructive suggestions.
Revision was made in the manuscript (line 33-35).
Minor comments
Comment 5: Line 30, square bracket: you should replace[1] [2] to [1-2]
Response
Thank you for constructive suggestions.
Revision was made in the manuscript (line 40).
Comment 6: All references need thorough change according to the: Instruction for Authors.
Response
Thank you for constructive suggestions.
Revision was made in references of the manuscript.

Reviewer 2 Report
The article is interesting and extends the knowledge already available in some aspect. However, I suggest that the authors broaden the description of the conditions in which they observed the described relationship. GDP per capita is an indicator with numerous limitations. It would be good to present the changes in Gini coefficient over the period of time studied. There is also no clear answer as to whether the legal regulations relating to environmental protection have changed radically over the years. It is not clear whether at the beginning of the 21st century there were radical changes in the way industrial investments were carried out, influenced by political decisions and regulations. Perhaps such a change was the cause of a reversal of the tendency identified in the research. The authors should clarify such doubts and consider also other limitations of the presented research much more broadly.
Author Response
Dear reviewer,
Thank you for providing the detailed and constructive suggestions for our manuscript. We have carefully reviewed, addressed each comment below and made corresponding changes in the manuscript (highlighted in yellow).
We apologize that the writing of the manuscript in the previous version was not clear and caused confusion. The English of the manuscript was edited by a professional company.
We hope you find the changes satisfactory.
Sincerely
Xiaomei Gan
Comment 1:
The article is interesting and extends the knowledge already available in some aspect. However, I suggest that the authors broaden the description of the conditions in which they observed the described relationship. GDP per capita is an indicator with numerous limitations. It would be good to present the changes in Gini coefficient over the period of time studied.
Response
Thank you for the constructive suggestions!
Although we tried our best to collect the data of GINI, However, some data cannot be obtained from authoritative platforms, especially in the early years. However, we added the urbanization ratio as another socio-economic indicator and made the regressions. Revision was made in the manuscript (line 183-185, line 205-208, Table2, Figure 3 and 5).
In addition, we took growth rates in log differences and carried out the instrumental variable estimation for robustness of our results. Revisions were shown in appendix 2and 3.
Comment 2:
There is also no clear answer as to whether the legal regulations relating to environmental protection have changed radically over the years. It is not clear whether at the beginning of the 21st century there were radical changes in the way industrial investments were carried out, influenced by political decisions and regulations . Perhaps such a change was the cause of a reversal of the tendency identified in the research. The authors should clarify such doubts and consider also other limitations of the presented research much more broadly.
Response
Thank you for the constructive suggestions!
Revision was made in the manuscript (line 321-325)

Reviewer 3 Report
Main comments:
Although the goal of the paper is clear, the originality of the paper among others must be justified. In additon, there are several points that need improvement. A more extensive literature review must be made linking the paper's subject/resutls with other studies. Furthermore, in order to explain the robustness of your results you must take growth rates in log differences due to the non stationarity of the variables. Additionally you must perform tests to exam for endogeneity and specifiacation errors.
Author Response
Dear reviewer,
Thank you for providing the detailed and constructive suggestions for our manuscript. We have carefully reviewed, addressed each comment below and made corresponding changes in the manuscript (highlighted in yellow).
We apologize that the writing of the manuscript in the previous version was not clear and caused confusion. The English of the manuscript was edited by a professional company.
We hope you find the changes satisfactory.
Sincerely
Xiaomei Gan
Comments 1
Although the goal of the paper is clear, the originality of the paper among others must be justified.
Response:
Thank you for the constructive suggestions!
Revision was made in the manuscript (line 67-75).
Comments 2
A more extensive literature review must be made linking the paper's subject/results with other studies.
Response:
Thank you for the constructive suggestions!
Revision was made in the manuscript (line 64-68).
Comments 3
Furthermore, in order to explain the robustness of your results you must take growth rates in log differences due to the non stationarity of the variables.
Response:
Thank you for the constructive suggestions!
We took growth rate of GDP per capital in log differences for the robustness of our results. Revision was shown in the appendix 2 and in the manuscript (line 135-137).
Comments 4
Additionally you must perform tests to exam for endogeneity and specification errors.
Response:
Thank you for the constructive suggestions!
We carried out a number checks for the robustness of the results and specification errors, such as standard two-way fixed effects estimation, exclusion of non-stationarity of variables, there steps for U-Curve hypothesis test ,periodical examination and so on. And we also conducted an instrumental variable estimation, which uses infant mortality as an instrument for GDP per capita, to simultaneously cope with endogeneity and panel non-stationarity. Revision was shown in the appendix 3 and in the manuscript ( line 151-157).

Round 2
Reviewer 1 Report
Dear Authors,
I appreciate your efforts in improving the article.
My question is why this time you didn't use official teamplate?
I have only two last suggestion.
Firstly: You should develop the conclusions. Your arcicle contain 28 pages (last version) and you write only 9 lines of conclusion!
Secondly: You should work about the References - text editing (font, italics, etc. - according to mdpi instruction for authors).
Best regards.
Author Response
Dear reviewer,
Thank you for providing constructive suggestions for our manuscript. We apologize that we did not fully understand your advice and we also apologize that the writing of the manuscript in the previous version was not clear .The English of the manuscript was checked again by a professional company.
We hope you find the changes satisfactory.
Best regards
Sincerely
Xiaomei Gan
Major comments
Comment 1: You should develop the conclusions. Your arcicle contain 28 pages (last version) and you write only 9 lines of conclusion!
Response
We apologize that we did not fully understand your advice.
The conclusion was revised as your suggestion. Revision was made in the manuscript ( line 331-348).
Comment 2: You should work about the References - text editing (font, italics, etc. - according to mdpi instruction for authors).
Response We apologize for misunderstanding your suggestion.
Revision was made in the manuscript according to instruction for authors (line 362-459.

Reviewer 3 Report
The authors have made all the revisions that I asked them for, and have illustrated them in new tables and discussion, and I thus recommend publication. The paper will, however, benefit from another round of proof-editing for the English language used, but I will not insist on this being a precondition for its publication.
Author Response
Dear reviewer,
Thank you for constructive suggestions for our manuscript. We apologize that the English edition of the manuscript in the round 1 was not satisfactory. The English of the manuscript was edited again by a professional company.
We hope you find the changes satisfactory.
Best regards
Sincerely
Xiaomei Gan
Comment
The authors have made all the revisions that I asked them for, and have illustrated them in new tables and discussion, and I thus recommend publication. The paper will, however, benefit from another round of proof-editing for the English language used, but I will not insist on this being a precondition for its publication.
Response:
Thank you for the constructive suggestion.
English of the manuscript was edited again by a professional company.
